# Tracking Traveling Ionospheric Disturbances through Doppler-shifted AM radio transmissions

Claire C. Trop[1,2], James LaBelle[2], Philip J. Erickson[3], Shun-Rong Zhang[3], David McGaw[2], Terrence Kovacs[2]

[1]Applied Physics Laboratory, Johns Hopkins University, Laurel, MD, USA
[2]Department of Physics and Astronomy, Dartmouth College, Hanover, NH, USA
[3]Haystack Observatory, Massachusetts Institute of Technology, Westford, MA USA

*Correspondence to*: Claire C. Trop (Claire.Trop@jhuapl.edu) or James W. LaBelle (james.w.labelle@dartmouth.edu)

**Abstract.** Six specialized radio receivers were developed to measure the Doppler shift of amplitude modulation (AM) broadcast radio carrier signals due to ionospheric effects. Five were deployed approximately on a circle at a one-hop distance from an 810-kHz clear-channel AM transmitter in Schenectady, New York, and the sixth was located close to the transmitter, providing a reference recording. Clear-channel AM signals from New York City and Connecticut were also received. The experiment confirmed detection of travelling ionospheric disturbances (TIDs) and measurement of their horizontal phase velocities through monitoring variations of the Doppler shift of reflected AM signals imparted by vertical motions of the ionosphere. Comparison of twelve events with simultaneous global navigation satellite system (GNSS) based TID measurements showed generally good agreement between the two techniques slightly more than half the time and substantial differences slightly less than half the time, with differences attributable to differing sensitivities of the techniques to wave altitude and characteristics within a complex wave environment. Detected TIDs had mostly southward phase velocities, and in 4 cases they were associated with auroral disturbances that could plausibly be their sources. A purely automated software technique for event detection and phase velocity measurement was developed and applied to one year of data, revealing that AM Doppler sounding is much more effective when using transmitter signals in the upper part of the AM band (above 1 MHz) and demonstrating that the AM Doppler technique has promise to scale to large numbers of receivers covering continent-wide spatial scales.

## 1 Introduction

Travelling Ionospheric Disturbances (TIDs) are a category of ionospheric variations characterized by propagating wavelike undulations in the ionospheric layer height and/or variations in ionospheric electron density. TIDs have been studied since the earliest days of ionospheric radio research [reviews by Hunsucker, 1982; Hocke and Schlegel, 1996]. They are typically classified as: medium scale (MSTIDs), having horizontal wavelengths of hundreds of kilometers, periods of 15-60 minutes, and horizontal phase speeds 250-1000 m/s; and large scale (LSTIDs), having horizontal wavelengths exceeding 1000 km,

periods of 30-180 minutes, and horizontal phase speeds of 400-1000 m/s [Hernández-Pajares et al., 2012; Otsuka et al, 2013; Tsugawa et al., 2004; Ding et al., 2014]. TIDs can be the ionospheric signature of atmospheric processes such as gravity waves [review by Yeh and Liu, 1974; Francis, 1975; Fritts, 1995] or can result from electrodynamically driven ionospheric processes

such as the Perkins instability [Perkins, 1973; Cosgrove and Tsunoda, 2002, 2004; Cosgrove, 2007; Kelley, 2011; Makela and Otsuka, 2011]. Atmospheric gravity waves associated with TIDs are an important mechanism of energy and momentum transport between atmospheric regions. For example, storm-time LSTIDs involve significant energy transport from high to low latitudes [Richmond, 1978; Yeh and Liu, 1974].  Gravity wave energy altitudinal transport, typically from lower to higher altitudes, has significant effects on atmospheric temperature dynamics, such as determining the seasonal variation of the

mesopause temperature minimum [e.g. She et al., 2022]. In addition to these important effects on atmospheric structure and energy/momentum flow, TIDs can significantly affect both trans-ionospheric and sub-ionospheric radio wave propagation, including impacting radio communications systems as discussed by e.g. Frissell et al. [2022].

TIDs have been detected by many remote sensing methods, including ionosondes, optical cameras, GNSS TEC, coherent and

incoherent scatter radars, and Doppler sounding using dedicated transmitters. From the earliest investigations, radio methods using transmitters of opportunity have played a significant role. Even before TIDs were first identified, variations in radio reflection height were detected by Essen [1935], using an early installation of US national time station WWV at 5 MHz frequency, and probably represent a TID detection. A host of early studies measured and statistically characterized TIDs with the Doppler sounding method using the regular and highly stable U.S. National timing standard signal WWV broadcast at HF

frequencies from Fort Collins, Colorado and received in Palo Alto, California [Chan and Villard, 1962; Davies et al., 1962; Fenwick and Villard, 1960; Sears, 1972]. Similar studies at HF frequencies exploited the Canadian timing signal CHU [Toman, 1975], an Australian telecommunications timing signal [Joyner and Butcher, 1980], and the Russian timing signal RWM [Reznychenko et al. 2020]. Tedd et al. [1984] observed one-hour timescale, ~5 degree variations in bearing deflections on 600-1000 km paths from transmitters of opportunity in central Europe, which matched TID-based modelling. In the 1980's, several

groups collaborated to measure the Japanese standard timing signal JYY on 8 MHz at multiple receivers, determining the horizontal wavelength for different frequencies in the TIDs and combining these to determine a dispersion relation which matches the theoretical prediction for waves just below the Brunt-Vaisäla frequency [Shibata et al. 1983; Tsuitsui et al., 1984].  Tanaka et al. [1984] used Doppler sounding of the JYY timing signal at multiple sites to investigate a mechanism of TID generation by a 1982 earthquake.

Probably the most sophisticated application of transmitters of opportunity to measure TIDs is Frequency Angular Sounding (FAS), whereby time series variations of Doppler shift, elevation of arrival, and azimuth of arrival can be inverted with certain assumptions to reconstruct the shape of the causative corrugations on the underside of the ionosphere and their time variations. Applying this technique to a transmitter of opportunity on a 700-km path using the large antenna array at the Kharkiv radio

telescope facility in Ukraine, Beley et al. [1995] obtained a remarkably detailed reconstruction of the underside of the

ionosphere (their Figure 7). Subsequent work showed that this method could work successfully with much smaller receiving systems observing a transmitter of opportunity such as CHU, suggesting that world-wide digisonde networks could be employed to do FAS on a regular basis [Galushko et al., 2003; Paznukhov et al., 2012].

Two recent papers illustrate TID detection using HF and MF transmitters of opportunity not related to national timing signals.

Chilcote et al. [2015] showed that observations of ~0.1-Hz Doppler shifts of carrier frequencies of two clear-channel AM radio stations over several hundred-kilometer paths at three receiver locations allow detection of TIDs. The study included one event with ~1-hour period simultaneously measured with GNSS TEC, coherent backscatter radar, and a nearby digital ionosonde (digisonde); during the event, a nearly 180-degree discrepancy was observed between the TID horizontal velocity inferred from spaced measurements of AM signals versus that inferred from time history of GNSS TEC maps. Recently,

Frissell et al. [2022] showed TID detection and characterization through statistics of large numbers of detected serendipitous links between amateur radio operators.

The work of Chilcote et al. [2015], and in particular the discrepant horizontal phase velocity measurement reported therein, inspires the study presented below. Compared to Chilcote's work, our study employs a larger number of receiver locations

distributed more widely around the signal source, and further develops methods of detecting and characterizing TIDs through Doppler sounding of AM radio stations using multiple receiver locations and advanced signal processing techniques. Section 2 describes the instrumentation used in this study and the places where it was deployed. Section 3 describes the methodology of TID detection and horizontal phase velocity determination. Section 4 discusses validation of these measurements using GNSS TEC data, and Section 5 shows evidence for auroral sources of some of the events. Section 6 applies automated data

analysis to a large amount of data suggesting how the method can scale up to much larger numbers of receiver locations. Finally, Section 7 summarizes the results and suggests future directions of study.

## 2. Instrumentation

The TID measurement system in this study consists of several elements: antenna/preamp, GPS antenna/receiver, upconverter, software defined radio, Raspberry-Pi microcomputer, and solid-state disk drives. The antenna is a vertically oriented square magnetic double loop 45 cm on a side, mounted on a ~4-foot mast. The resulting antenna pattern is a dipole

with a null at zero elevation in one direction, with capability of receiving AM signals down to relatively low elevations in almost all directions. An active pre-amplifier mounted on the antenna mast amplifies, band-limits, and buffers the induced signal to transmit it down 100-500 feet of coaxial cable (depending on site). Figure 1a shows the antenna/preamplifier in a typical setting on the roof of an academic building. At the receiver end is a bias-tee allowing pre-amplifier power and RF

signal to be sent on the same cable. In the receiver, the signal is upshifted by 129.6 MHz by mixing with a GPS-disciplined

local oscillator signal generated in custom electronics, putting the AM broadcast band in the sampling range of the software

defined radio (SDR). The RTL-SDR (using the RealTek chipset) uses a GPS-disciplined local oscillator synthesized in

custom electronics and together with associated processing software running on a Raspberry-Pi microcomputer to record in-

phase and quadrature waveforms corresponding to a 1 MHz bandwidth centered on 1100 kHz, covering most of the AM

band (600-1600 kHz). Data collection runs continuously for 14-19 hours/day, centered on nighttime. Figure 1b shows the

receiver with attached cables and disk drive, and Figure 1c is a block diagram of the receiving system.

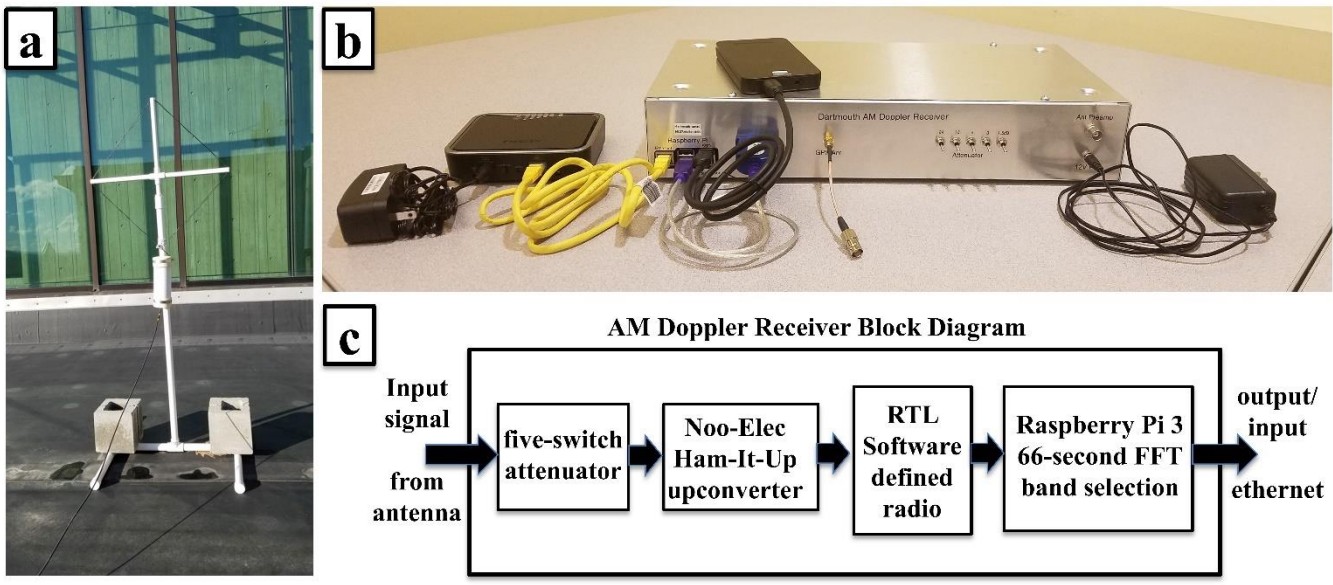

Figure 1: a) The antenna/preamplifier used in the AM Doppler receiving systems, as deployed on the rooftop of an academic

building; b) the AM Doppler receiver with associated cabling and external disk drive; c) a block diagram of receiver.

Initial data processing is performed in real time on the Raspberry-Pi. Achieving multiple narrow effective bandwidths for

AM carrier detection from the input continuous I/Q sample stream requires a very long (64M) Fourier transform. To

accomplish this task with limited computer memory, an 8192x8192 array implementation is used which is a variation on

those described extensively in the literature [e.g., Hocking, 1989; Ransom et al., 2002; and references therein]. This

implementation has the advantage not only of limiting memory requirement, but the computation can be limited to the

desired frequency resolution.  In this case, approximately one hundred 8-Hz bands are used bracketing the frequencies of

each AM signal, which in the US are licensed on 10 kHz spacings within the measured 1-MHz band. The resulting ~100

power spectra have 0.016 Hz frequency resolution and a cadence of 33.5 s. Operation during 12-14 nighttime hours yields

160 MB of compressed data per day. From most sites, the data are transmitted daily to a master server at Dartmouth College,

where custom software produces daily survey spectrograms displaying signal variation on selected AM stations. At sites having limited data transmission, software on the local Raspberry-Pi produces the survey spectrograms which are transmitted daily as compact PDF files. At those sites, locally stored data are periodically backed up to an external disk drive which is mailed to Dartmouth.

From April 2020 through March 2021, the hardware and software systems described above were deployed at six locations in the northeastern United States: Hanover, New Hampshire; New Bedford, Massachusetts; Ithaca, Schenectady, and Massena, New York; and Jenny Jump, New Jersey. Figure 2 shows a map of these locations and Table 1 gives their exact latitudes and longitudes. The stations are arrayed around a targeted clear-channel and high power (50 kW) AM radio transmitter, 810 kHz WGY in Schenectady New York, indicated by a black star in Figure 2. Under FCC license regulations, clear-channel AM

radio stations are the only allowed commercial transmission in a given frequency range and in theory should not be subject to interference. Red square symbols in Figure 2 show the nominal reflection locations of the signal paths of WGY to five of the receivers assuming one-hop propagation. The nominally measured reflection points cover a clock-like pattern spanning an approximate 200 by 200 km area. Separation distances between transmitters and receivers must be kept within an optimal range to avoid dominance of ground wave at short baselines and dominance of multi-hop propagation at longer baselines.

Receiver separations of ~200 km in this experiment were motivated by the success of previous investigations [e.g., Chilcote et al., 2015]. Some of the data shown below pertains to additional clear-channel AM stations: 1560 kHz WFME in New York City and 1080 kHz WTIC in Hartford, Connecticut, marked by blue and green stars, respectively. Blue and green triangles in Figure 2 show the nominal reflection locations of the signal paths of WFME and WTIC to the six receivers, which cover a similar area of approximately 200 by 200 kms each, but offset from that covered by the paths to the primary

targeted AM station WGY. As pointed out by Chilcote et al. [2015], monitoring TIDs via Doppler sounding of clear-channel AM signals requires placing one receiver next to the target AM transmitter in order to record a reference signal, allowing separation of ionospheric Doppler shifts from variations in carrier frequency due to imperfect transmission equipment. In the present experiment, this is achieved with the receiver at Schenectady in the case of receiving WGY, the receiver at Jenny Jump in the case of receiving WTIC, and the receiver at New Bedford in the case of receiving WFME. In practice, only the

WGY signal measured by the receiver in Schenectady required corrections to remove AM transmitter carrier drift.

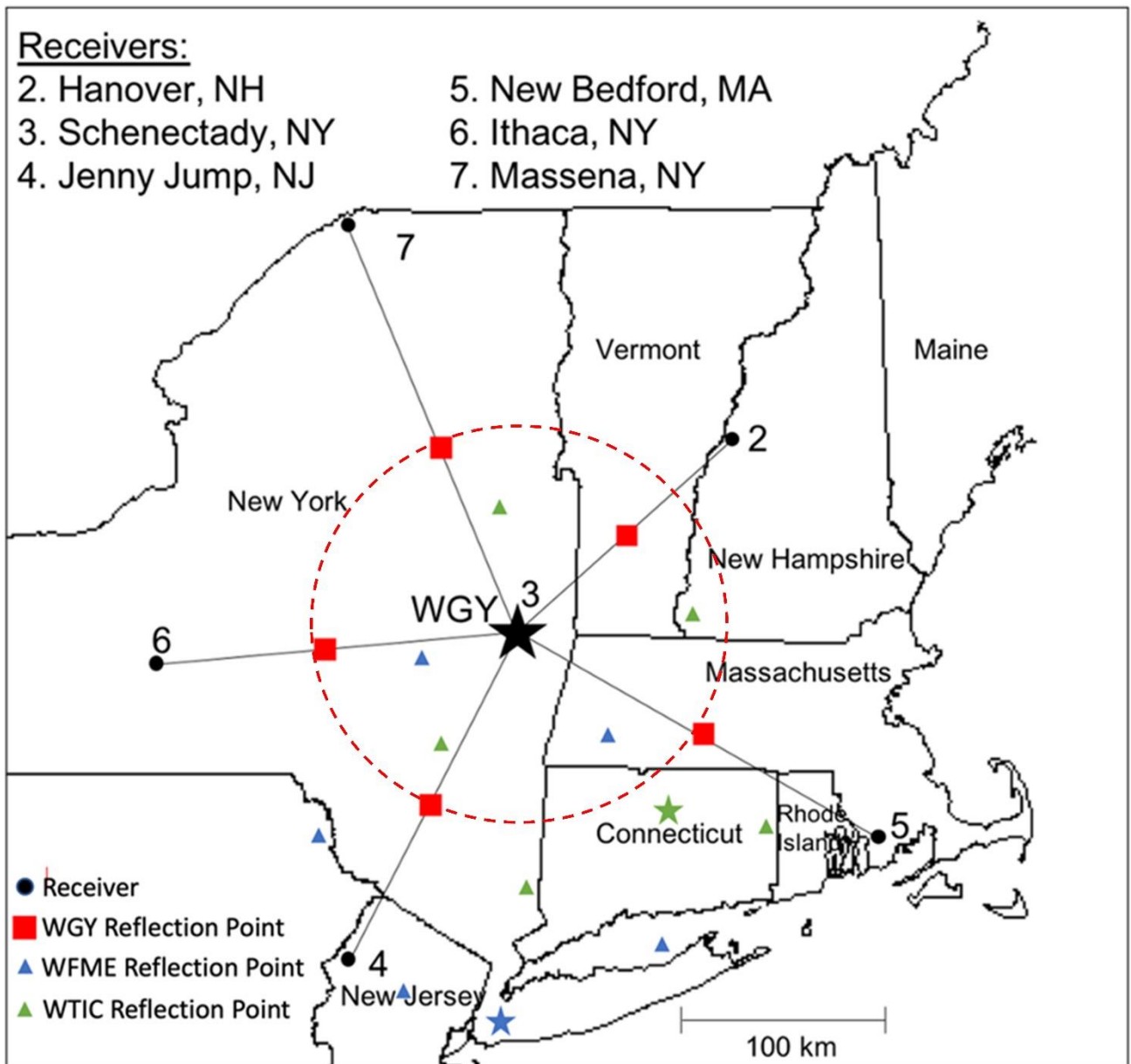

Figure 2: Map showing locations of transmitters (stars) and receivers (circles). Assuming specular, single-hop reflection places
reflection points (squares and triangles) at the midway points. The network is designed for reception of the 810 kHz WGY
clear-channel AM transmitter out of Schenectady, NY. The receivers encircle the WGY transmitter measuring the reflected
wave in all directions over similar path lengths. Additional clear channel signals at 1080 kHz (WTIC) and 1560 kHz (WFME)
can be detected by the network, but reflected signals are measured over a limited angular extent across different path lengths.

Table 1: Locations of AM Doppler sounder receivers used in this study.

| Receiver | Latitude (°N) | Longitude (°E) |
|---|---|---|
| Hanover, NH | 43.705 | -72.286 |
| Schenectady, NY | 42.813 | -73.912 |
| Jenny Jump, NJ | 40.907 | -74.925 |
| New Bedford, MA | 41,595 | -70.910 |
| Ithaca, NY | 42.496 | -76.431 |
| Massena, NY | 44.924 | -74.898 |

## 3. Methodology

Figure 3 shows the reflected skywave from the 810 kHz WGY clear-channel station from each receiver from 2100 to 1200 UT on the night of 26 September, 2020. Evidence of a TID passing over the receiver array is apparent in the coherent variations in the signal attributed to the Doppler shift from 0130 to 0400 UTC (2030-2300 LT) on all six receivers. The

signatures strongly resemble those reported by Chilcote et al. [2015] at 2330-0330 UTC (1830-2230 LT) on April 12, 2012, although the earlier measurement involved only three receivers.

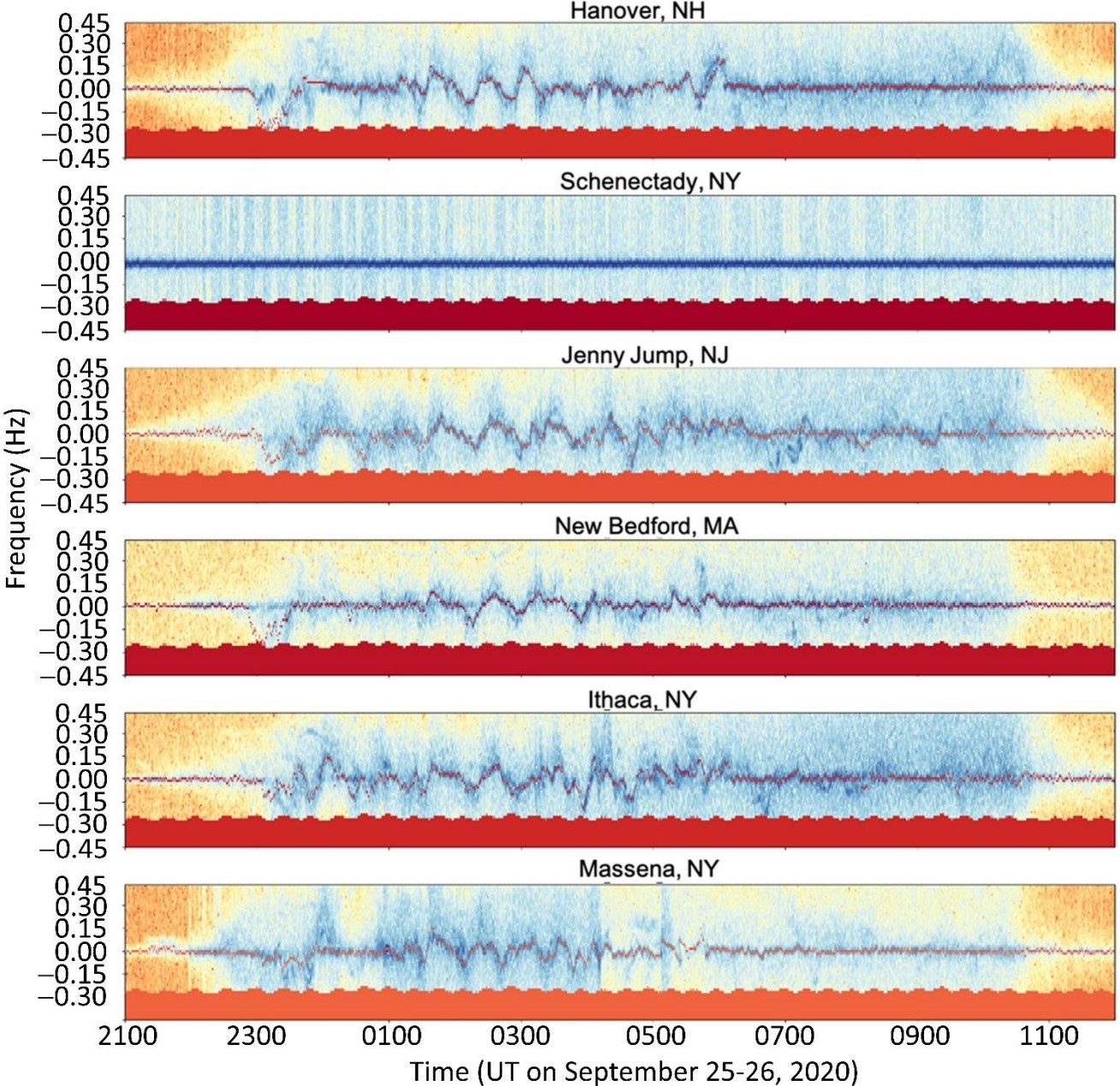

Figure 3: Spectrograms of the 810 kHz signal received at each location in the network covering 0.9 Hz bandwidth from 2200 UT on 25 September to 1200 UTC on 26 September, 2020. The solid-colored blocks at the base of each spectrogram illustrate the corrections made to account for variations in the frequency of the transmitted signal. The overlaid (red) trace is the extracted waveform generated by the semi-automated tracking algorithm with manual correction, as described in the text.

The red dashed lines on each receiver in Figure 3 trace the semi-automated tracking of the waveform used for analysis of TID characteristics. Semi-automated tracking software starts with a manual operation: through inspection of the frequency-time diagram, the user identifies times when the offset frequency of the carrier signal is clear and makes a large excursion from its nominal frequency. The software then follows the signal backward and forward from these manually determined set points by identifying the frequency of the maximum signal in preceding or following time intervals, with the step size in time defined by the resolution of the measurement (30 s) and the step size in frequency capped to prevent the tracking from being discontinuous. Upon completion of this preliminary tracking, the software displays the determined preliminary track superposed on top of the spectrogram, allowing the user to identify intervals when the preliminary track diverges from the skywave. The software allows the user to insert time and frequency coordinates that override the automated determinations and update the tracking. Challenges to the automated tracking include the presence of a high intensity ground wave (for example, see the New Bedford receiver at 0400 UTC), multi-path conditions resulting in multiple Doppler shifted signals at different frequencies (for example, see the Jenny Jump receiver at 0300 UTC), or events when the signal is multi-valued and folds back on itself (third panel of Figure 6 of Chilcote et al. [2015]). The manual correction ensures a consistent carrier trace across all receivers.

A set of traces from a ring of receivers can be used to estimate the horizontal phase velocity of the TIDs detected by the system. The time delay of the variations of the Doppler shifted signals between pairs of stations is the measured quantity used to determine the phase velocity. This time delay is determined using cross correlation between each receiver's signal and the Hanover receiver's signal. A high correlation coefficient identifies maximum similarity between the signals. Using a sliding window, the cross-correlation function is executed across the entire domain of traced signals, and time intervals with stationary time delay and a high correlation coefficient are identified as intervals potentially containing TIDs. Stationarity, defined here as a stable delay that does not change with time for an extended period, typically of order one hour, is required to ensure that the disturbance causing the Doppler shifts is consistent in shape and velocity across the entire array. Figure 4 shows the stationary, high correlation time delays calculated with respect to the Hanover (light green) receiver for the event on the night of 26 September, 2020 between 0130 and 0200 UTC. From the given time delays, it is immediately evident that the observed TID is propagating in the S/SE direction across the array as indicated by the red arrow.

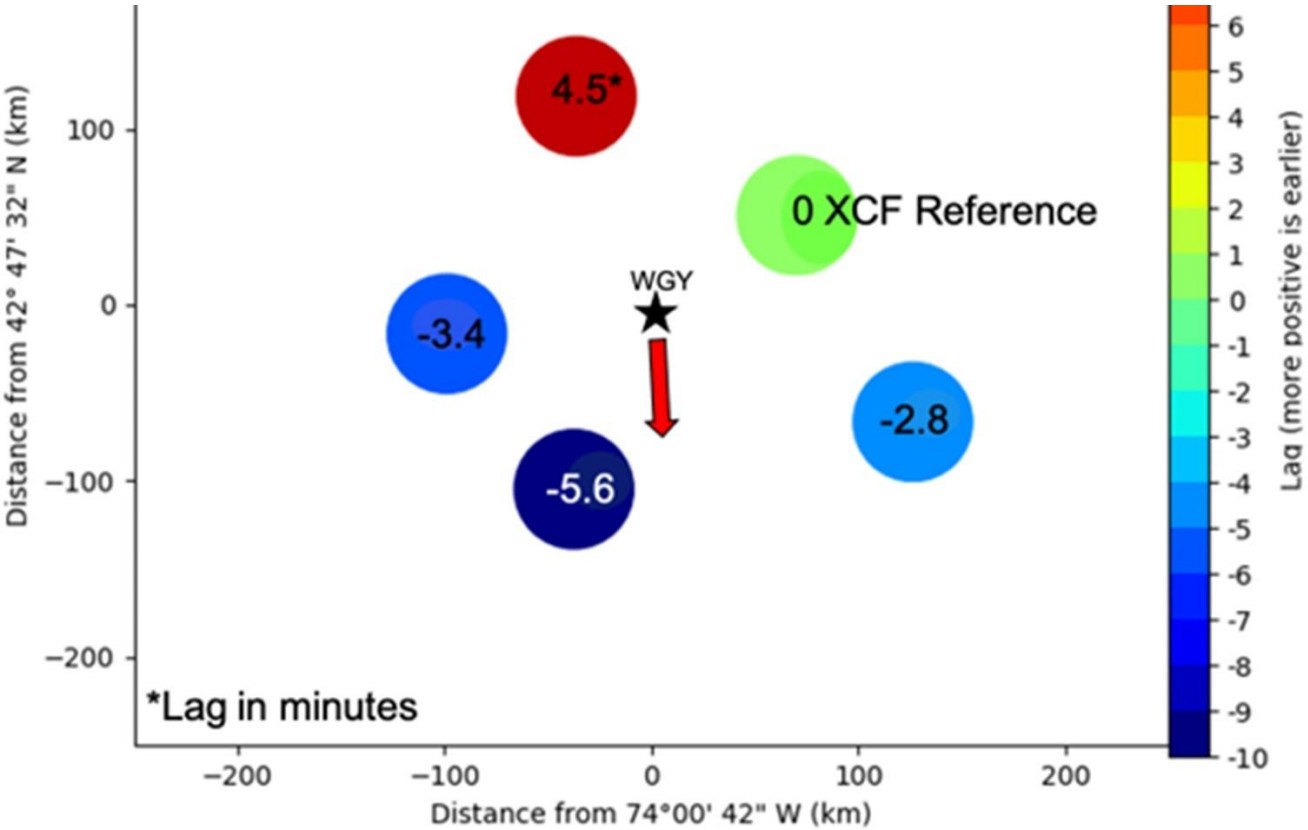

Figure 4: Map showing the location of each reflection point along with the time lag between frequency variations detected at each receiver and those detected at the Hanover, NH receiver (arbitrarily chosen as reference), for the event detected at 0200 UTC on 26 September, 2020. Red colors indicate positive lags (signals arriving earlier), and blue colors indicate negative lags (signals arriving later). The observed lag times indicate a southern direction of propagation (arrow) for the observed TID.

We use three methods for determining the phase velocity from the time delays, denoted as triad, slowness, and sine fit method, respectively. The triad method is a vector geometry approach to phase velocity which can be statistically analyzed because our experiment includes many baselines; i.e. many combinations of receiver triplets. A phase velocity can be calculated for each triplet as follows: for two pairs of receivers composing the triplet, the distance between reflection points is divided by the time delay between the received signals and inverted to get vector components, and adding these vectors provides an estimate of inverse phase velocity pertinent to the triplet. This calculation is repeated for all possible combinations of three receivers and averaged to get a final estimate for the phase velocity:

$$V_{phase} = \frac{N}{\sum_{n=1}^{N!/3!(N-3)!}\frac{2}{3}*\left[\sum_{i=1}^{3}\sum_{j=i+1}^{3}\frac{\vec{lag}_{ij}}{d_{ij}}\right]}, \tag{1}$$

The "sine method" takes advantage of the arrangement of the reflection points, which lie approximately on a circle of radius R=100 km centered on the Schenectady transmitter, indicted by a dashed line in Figure 1. Using a Cartesian coordinate system with origin at Schenectady and with x directed eastward and y directed northward, this circle is described by $x^2+y^2=R^2$. Suppose a plane wave approaches this circle of receivers from the north, propagating in the −y direction with velocity V. If the plane wave reaches the top of the circle (x=0, y=R) at time t=0, the equation for a wave front is y=R−Vt. Defining θ as the around the circle angle off of the y-axis, x-values of receivers around the circle are given by x=R sinθ. Inserting these x and y expressions into the equation for the circle and solving for t yields

$$t = (R/V)(1- \cos\theta) \tag{2}$$

The time delay as a function of receiver position θ is therefore a sine wave; in this case the receiver having minimum delay is that at θ=0 (at the northern edge of the circular array of receivers), and the receiver having maximum delay is that at θ=π (at the southern edge of the array). To treat plane waves coming in at other angles requires merely rotating the entire problem by the appropriate angle; for a plane wave directed at an angle ϕ off of the y-axis, the time delay as a function of angular receiver position around the array would be:

$$t = (R/V)(1 - \cos(\theta - \phi)) \tag{3}$$

This function is fit to the measured delays of the five receivers arrayed at known angles θ around the origin (Schenectady) to determine the angle ϕ and the magnitude R/V, from which the direction of propagation and the horizontal phase velocity of the plane wave can be determined, since R is known.

The slowness method for tracking activity captured by a receiver array was originally developed in the seismology community [Lacross et al., 1969] and also applied to location of thunder sources in storms [Johnson et al., 2011]. Chum et al. [2014] and Chum and Podolska [2021] adapted this method for TIDs with a Doppler sounding array. As implemented with our array, the slowness parameter is defined as the inverted phase velocity. Sampling a range of x and y components of the slowness generates an energy map, as defined by Chum et al. [2014], which is a measure of the alignment of the waveforms at the center of the array. The energy map, $W(s_x, s_y)$, is defined by:

$$W\left(s_x, s_y\right) \propto \sum_{t_i=-\frac{\Delta t}{2}}^{\frac{\Delta t}{2}} \left[ \frac{\sum_{n=1}^N f_{Dn}(t_i + s_x \Delta x_n + s_y \Delta y_n + s_z \Delta z_n)}{N} \right]^2 , \tag{4}$$

and normalized according to:

$$C\left(s_x, s_y\right) \propto \frac{W(s_x,s_y)}{\frac{1}{N} * \sum_{n=1}^N \left[ \sum_{t_i=\frac{\Delta t}{2}}^{\frac{\Delta t}{2}} f_{Dn}(t_i)^2 \right]} , \tag{5}$$

where $s_x$ and $s_y$ are the x- and y-components of the slowness velocity, and $f_{Dn}$ is the Doppler shift measured at time $t_i$ at the location of each reflection point. The $\Delta x_i$ values in equation (4) are the distance from each reflection point to the center of the

245 array. Since the array uses only the 810 kHz WGY signal for this study's analysis, the altitudes of reflection are assumed to be equal and the $\Delta z$ term can be neglected. Figure 5 shows the energy map calculated using the slowness algorithm for 0130-0200 UTC on the night of 26 September, 2020. We define the peak of the energy map (black circle) as the top 97% of values, called peak slowness correlation coefficients. The distance from the center of the map to the centroid of the peak slowness correlation coefficients is the estimate of the magnitude of the inverse phase velocity, and the azimuthal direction of that

centroid relative to the center of the map is the estimate of the direction of propagation. The inverse phase velocities associated with the extreme points of the region of peak slowness correlation coefficients can be taken as estimates of the relative uncertainty in magnitude (based on the inner and outer extrema relative to the origin) and direction (based on the azimuthal extrema). These relative uncertainty values are useful in comparing one phase velocity to another calculated with the slowness method, but they bear no obvious relation to the absolute uncertainty of the measurement.

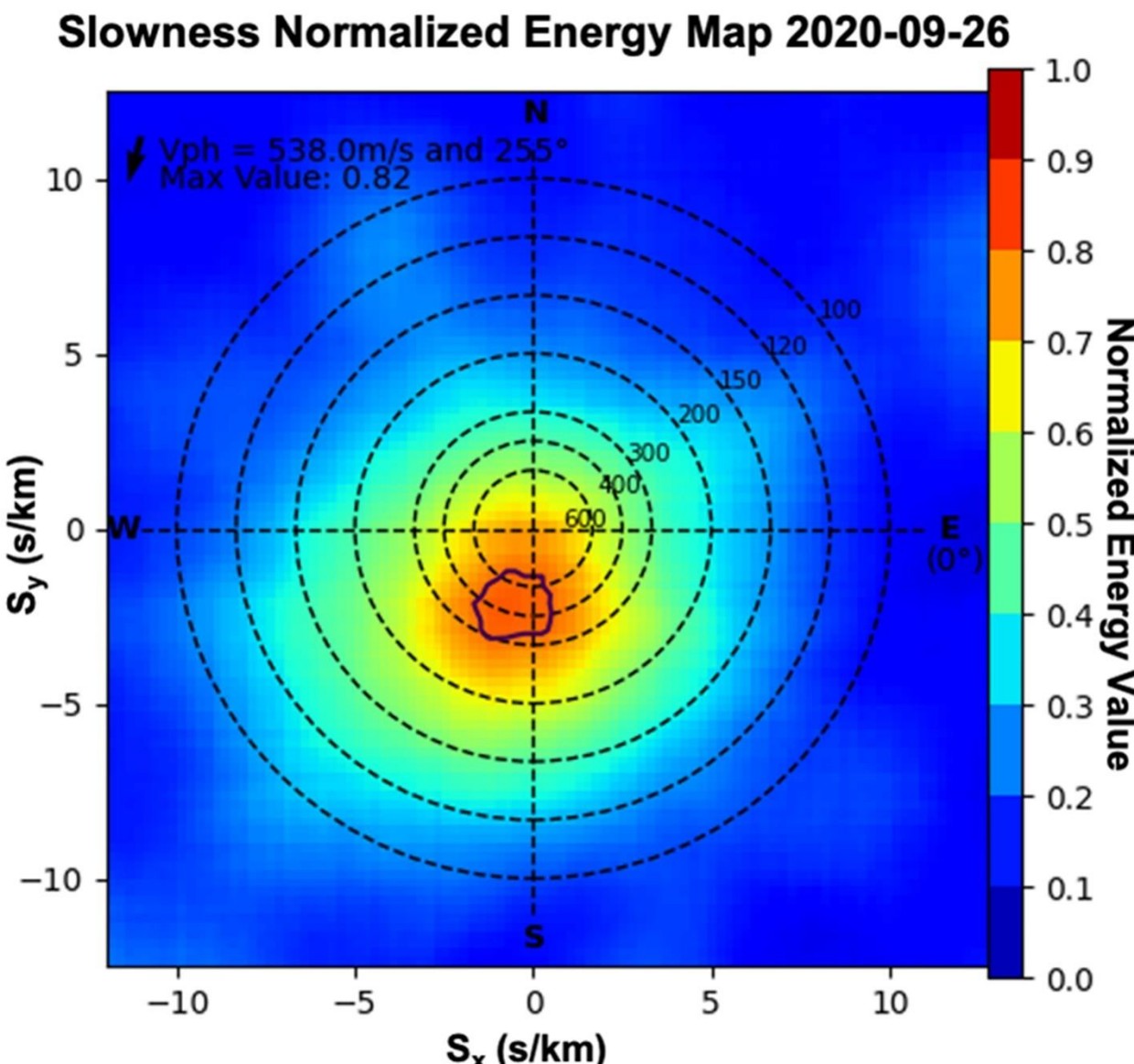

Figure 5: The slowness phase velocity energy map for the event occurring at 0200 UTC on 26 September, 2020. The region outlined with a solid trace indicates the peak region (top 3% of slowness correlation coefficients). The phase velocity magnitude is calculated from a weighted centroid of the energy values within the peak region, and the angle to that centroid indicates the direction of propagation.

Table 2 lists the events detected between 4 February, 2020, and 31 March, 2021, using the Schenectady station (WGY, 810 kHz) as the transmitter. These events have been initially identified by manual inspection of reflected skywave plots similar to those shown in Figure 3. The semi-automated tracking described above has been applied to them, and their phase velocities have been calculated using all three methods and results tabulated in the table.

The triad and sine-fit methods both depend on the time delays determined from the normalized cross-correlation, making them especially sensitive to tails/outliers in the distribution of those delay times. Furthermore, both of these methods, but especially the sine method, become significantly less effective if fewer than five receivers are available. For these reasons, the slowness method, potentially combined with the normalized sliding cross correlation for a consistency check, has the best potential for automation and scaling up these methods for tracking TIDs with large numbers of receivers. Velocities determined from the sine and triad method methods generally agree with those from the slowness method to within 45 degrees with one (triad) to two (sine) outliers within 90° for the direction of propagation. The triad has a median of 343 m/s, mean of 394 m/s, and standard deviation of 232 m/s. The sine has a median of 298 m/s, mean of 361 m/s, and standard deviation of 258 m/s, but is only functional when all five receivers were operational (9 out of 13 events).

Focusing on the slowness method results in Table 2, there are seven TIDs detected traveling in the S/SE direction, three events in the SW direction, and two events in the NW direction. For the event on 10 December, 2020, the slowness method indicates an infinite phase velocity (i.e. the peak correlation coefficients region encircles the origin), although the sine and triad methods imply a southwesterly direction. Of the thirteen events, four have phase velocities between 100-300 m/s, three between 300-400 m/s, two between 400-500 m/s, and four above 500 m/s. Larger phase velocities are accompanied by greater uncertainty. The four events below 300 m/s are likely classifiable as MSTIDs while the remaining eight events above 300 m/s are likely LSTIDs. The median velocity is 347 m/s, the mean velocity was 411 m/s, and the standard deviation is 199 m/s. For reference, the principal TID event analyzed by Chilcote et al. [2015] has a phase velocity of 330 m/s propagating in the northeast direction.

Table 2: List of events detected using WGY Schenectady (810 kHz).

| Event Date + Time | Slowness Phase Velocity Magnitude (Direction) | Triad Method Velocity Magnitude (Direction) | Sine Method Velocity Magnitude (Direction) | Number of Receivers |
|---|---|---|---|---|
| 2020-03-23: 0400 UTC | $478^{+2277}_{-260}\ m/s$ (S) | 414 (SSE) | n/a | 3 |
| 2020-05-06: 0400 UTC | $673^{+770}_{-373}\ m/s$ (SE) | 361 (SE) | 352 (SE) | 4 |
| 2020-07-14: 0530 UTC | $358^{+88}_{-124}\ m/s$ (SSE) | 305 (SE) | 298 (SE) | 5 |
| 2020-09-09: 0400 UTC | $150^{+26}_{-8}\ m/s$ (NW) | 237 (N) | 248 (NE) | 5 |
| 2020-09-22: 0500 UTC | $267^{+126}_{-85}\ m/s$ (SE) | 343 (SE) | n/a | 4 |
| 2020-09-26: 0200 UTC | $538^{+218}_{-244}\ m/s$ (S) | 428 (S) | n/a | 5 |
| 2020-09-27: 0400 UTC | $335^{+66}_{-97}\ m/s$ (SE) | 300 (SE) | 304 (SE) | 5 |
| 2020-09-29: 0200 UTC | $859^{+218}_{-425}\ m/s$ (SW) | 1099 (S) | 1026 (SSW) | 5 |
| 2020-11-15: 0730 UTC | $332^{+107}_{-84}\ m/s$ (SW) | 446 (W) | n/a | 5 |
| 2020-12-10: 0700 UTC | n/a | 498 (SSW) | 381 (SW) | 5 |
| 2020-12-29: 0700 UTC | $229^{+119}_{-34}\ m/s$ (NW) | 158 (W) | 155 (WNW) | 4 |
| 2020-12-30: 0700 UTC | $418^{+168}_{-155}\ m/s$ (SW) | 225 (NW) | 219 (NW) | 5 |
| 2021-01-26: 0230 UTC | $299^{+192}_{-112}\ m/s$ (S) | 313 (SE) | 270 (SE) | 5 |

.

Assessing the possibility that E-region reflections might affect the results in Table 2 motivates examination of vertical ionograms at 7.5-minute cadence from mid-latitude observations at Westford, MA (42.6 N latitude / 288.5 E longitude). The ionosonde data, comprising station MHJ45 within the Global Ionospheric Radio Observatory (GIRO), originated from the University of Massachusetts Lowell's ionosonde station operated on the MIT Haystack observatory grounds. MHJ45 was located on the eastern edge of the study region. No sporadic E was recorded during any of the TID events in Tables 2-3 of our paper. Around event times, Sporadic E was detected about an hour before one of the events (on Sept 26, 2020) and an extremely weak Sporadic E layer was detected at 2 MHz frequency for 90 minutes after one of the events (December 30, 2020). Based on this survey, the results in Table 2 are not significantly impacted by sporadic E propagation.

**4. Validation with GNSS Technique**

For validation beyond the AM network observations, detected TID events can be compared with simultaneous observations of TID characteristics obtained from global navigation satellite system (GNSS) total electron content (TEC) data. TEC is the

integrated electron density along a path between a ground receiver and a satellite. Time delays between different frequencies along the same path allow TEC to be calculated (Cooper et al., 2019)**.** The measured TEC quantity is then projected

geometrically onto the vertical axis and detrended from the baseline electron content. Detrended TEC as a function of time and space can show periodic increases and decreases which are linked to TID activity (Tsugawa et al., 2007, Zhang et al., 2022). The TEC technique and the AM doppler sounding technique have different sensitivities in wave altitude and wave characteristics.

The detrended TEC (or differential TEC, dTEC) data, plotted in keogram form, is generated using global GNSS observations processed by MIT Haystack Observatory as part of operations for the Millstone Hill Geospace Facility (e.g. Zhang et al., 2017; Zhang et al., 2018). In this analysis, 30-minute and 60-minute sliding windows are used to determine the baseline electron content. The resulting keograms are analogous to slit cameras. A range of latitudes or longitudes (y-axis) is chosen and the dTEC for that narrow range is calculated and plotted for a number of hours (x-axis). The TIDs become visible on the

keograms as coherent regions of periodic increases and decreases in TEC which advance along the diagonal (changing latitude or longitude with time). GNSS data have been analyzed for the time intervals of the TIDs detected by the AM Doppler receivers (listed in Table 2). Figure 6 shows the 35 to 55°N latitude keogram along the selected slit from -75 to -70°E longitude, for the TID event on 26 September, 2020, using a 30-minute detrending window. From the angle of the coherent wave structure, the calculated phase velocity propagated southward at 430 m/s around 0200 UTC. This phase

velocity from GNSS TEC agrees well in magnitude and direction with that determined by the AM Doppler receiver network on the same data and time (see Table 2).

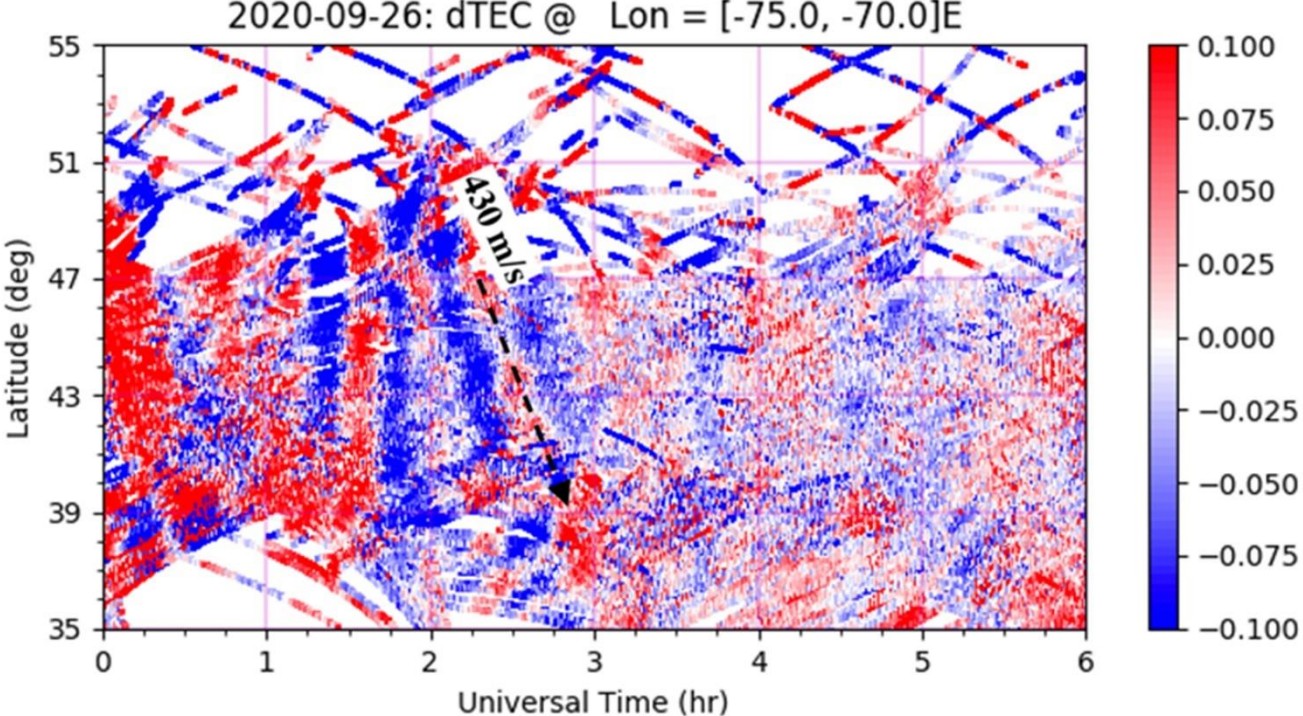

Figure 6: Differential TEC keogram for a TID detected via GNSS on the night of September 26[th], 2020, using a 30-minute detrending window. The keogram indicates a coherent structure propagating southward with a phase velocity of 430 m/s. The keogram corroborates the TID event direction and speed measured with the AM Doppler receiving network at the same time.

Table 3 presents results of comparing horizontal phase velocities of each event identified by AM Doppler sounding, estimated by the slowness method (second column) and estimated from simultaneous keograms from GNSS-TEC data using two different detrending windows, 30 minutes (third column) and 60 minutes (fourth column). In seven of twelve events for which an estimate is possible using both detrending windows, there is a significant difference between the magnitudes of the phase velocities inferred from the GNSS-TEC data; in five of these cases the value estimated with the 60-minute detrend window agrees better with that inferred from the AM Doppler method, and in two cases the value estimated with the 30-minute detrending window agrees better.

 In all but three cases for which a coherent TID signature was observed in the GNSS-TEC data, the directions of propagation determined from at least one of the GNSS-TEC calculations is within 45 degrees of those determined from the AM Doppler receiver network/slowness analysis. One exception occurs on 01-26-2021 when a 90-degree discrepancy is observed. The

other exceptions are 180-degree discrepancies occurring on 30-12-2020 and 09-22-2020; in the first case, for the GNSS-TEC estimate only the 30-minute detrend window gives a result, and in the second case the triad and sine methods give a direction opposite to that of the slowness method and in agreement with the GNSS-TEC estimate. For the event on 26 January, 2021, the AM Doppler system detects a TID which propagates southward with a phase velocity 219 m/s and has been linked to an auroral source (see Section 5), and the TEC keogram detects an event propagating westward with a phase velocity of 130 m/s. It is likely in this case that the two techniques detect two distinct TIDs, one caused by an auroral event and prominent at lower altitudes and one potentially caused by the electrodynamically driven Perkins instability (Perkins, 1973; Narayanan et al., 2018). The configuration is suggested by the TID's direction of propagation and phase speed which would have more prominent effect at the higher altitudes to which the GNSS TEC technique is more sensitive.

Concerning magnitudes of the horizontal phase velocities, there is somewhat less concordance between AM Doppler and GNSS-based estimates. In seven of twelve cases, the magnitudes agree to within 150 m/s (ranging from 15 m/s to 136 m/s), although in one of these cases, the direction differs by ninety degrees. In five of twelve cases, the magnitudes disagree by more than 150 m/s (ranging from 157 m/s to 366 m/s). In one of these cases the directions agree, but in the others the directions disagree by 90 degrees or more. In the four cases for which the GNSS-TEC method indicates only a meridional or zonal component, three of these are within twenty percent of the phase velocities determined from the AM Doppler receiver network/slowness analysis. For the other five events, the GNSS-TEC method indicates both meridional and zonal components. Since phase velocities must be inverted to be added, the smaller amplitude component becomes the dominant phase velocity. In these five cases, the slowness calculated phase velocity is larger than that inferred from the GNSS-TEC method. In fact, out of the eleven events for which the directions agree between the two techniques, nine had slowness phase velocities greater than their TEC counterparts.

Table 3: Comparisons of TID events detected with both AM Doppler sounding and GNSS.

| Event Date and Time | Slowness Phase Velocity Magnitude (Direction) | GNSS TEC Phase Velocity, 30-min detrend Magnitude (Direction) | GNSS TEC Phase Velocity, 60-min detrend Magnitude (Direction) | Kp index |
|---|---|---|---|---|
| 2020-03-23: 0400 UTC | 478 m/s (S) | 485 m/s (S) | 517 m/s (S) | 3+ |
| 2020-05-06: 0400 UTC | 673 m/s (SE) | 320 m/s (S) and 190 m/s (E) = 163 m/s | 537 m/s (S) | 2+ |
| 2020-07-14: 0530 UTC | 358 m/s (SSE) | 645 m/s (S) and 165 m/s (E) = 160 m/s | 645 m/s (S) and 384 m/s (W) = 330 m/s | 4- |
| 2020-09-09: 0400 UTC | 150 m/s (NW) | 170 m/s (N) 150 m/s (W) = 113 m/s | 129 m/s (N) and 160 m/s (W) = 101 m/s | 0 |
| 2020-09-22: 0500 UTC | 267 m/s (SE) | 154m/s (N) 155 m/s (W) = 109 m/s | No pattern observed | 3− |
| 2020-09-26: 0200 UTC | 538 m/s (S) | 430 m/s (S) | 445 m/s (S) | 4 |
| 2020-09-27: 0400 UTC | 335 m/s (SE) | 140 m/s (W) [well-defined], 115m/s (S) [faint] | 320 m/s (S) | 4 |
| 2020-09-29: 0200 UTC | 859 m/s (SW) | 580 m/s (S); 645 m/s (S) = 430 m/s | 780 m/s (S) | 4 |
| 2020-11-15: 0730 UTC | 332 m/s (SW) | 130 m/s (N); 165 m/s (W) = 102 m/s | 85 m/s (S) and 60 m/s (W) = 50 m/s | 0 |
| 2020-12-10: 0700 UTC | n/a | 80 m/s (S) and 60 m/s (W) (multiple wavefronts) | 103 m/s (N) and 60 m/s (W) = 55 m/s | 1 |
| 2020-12-29: 0700 UTC | 229 m/s (NW) | 215 m/s (S) and 60 m/s (W) = 58 m/s, also weak northward | 129 m/s (N) and 60 m/s (W) = 55 m/s | 1- |
| 2020-12-30: 0700 UTC | 418 m/s (SW) | 145 m/s (N) and 80 m/s (W) = 70 m/s (multiple wavefronts) | 107 m/s (N) and 60 m/s (W) = 52 m/s | 2- |
| 2021-01-26: 0230 UTC | 299 m/s (S) | 130 m/s (W) (multiple waves) | 120 m/s (W) | 4- |

A possible explanation for this observed discrepancy is that the GNSS-TEC method, relying on an integral of electron density through the ionosphere, is most sensitive in general to the peak electron density regions at higher F2 altitudes in the ionosphere, while the AM Doppler sounding technique exclusively looks at reflection off the bottom of the ionosphere. Since a TID has a vertical wavenumber, its characteristics are altitude dependent. The Doppler sounding technique is only able to determine information about the horizontal wavenumber, but contributions from the vertical may be sufficient to

explain the discrepancy in magnitude between the two techniques. Another potential factor to consider is the detrending

technique employed in GNSS TID analysis, which may lead to the detection of only specific components of multifrequency waves. In the context of the 30-minute and 60-minute sliding windows used in this study, larger scale and longer periodicity waves tend to be attenuated. Table 3 appears to indicate that these GNSS-TEC derived TID phase speeds are indeed slower, and therefore these waves likely contain some MSTID components.

In aggregate, the TEC and AM Doppler observations show that the TID wave environment is sometimes complex, with multiple waves of different characteristics simultaneously present and detected favorably by one technique or the other. The

405 consistency of the two techniques for both magnitude and direction of the horizontal phase velocity for a slight majority of the observed events in this work suggests that they often observe the same TIDs. However, the occurrence of discrepancies almost half the time, including cases in which the two methods give opposite directions, suggests that the unexplained discrepancy between directions of propagation previously seen in Chilcote et al. [2015] was not uncommon and results from a complex wave environment, similar to the 15 November, 2020, and 26 January, 2021, events in this study. Possibly, a

410 contributing factor was additional error in the AM Doppler technique introduced by their experimental set up which relied on multiple frequencies from different radio transmitters and only used three receivers

**5. Connections to Auroral Activity**

Auroral geomagnetic storms and substorms can enhance the auroral electrojet, which then initiates TIDs through Lorentz forces and Joule heating (Ding et al., 2008). To detect the occurrence of this mechanism, electrojet enhancements can be tracked through the Auroral Electrojet (AE) Index (Davis and Sugiura, 1966) and spatially and temporally isolated using magnetometer data from instruments maintained in several arrays of arctic observatories, such as the Canadian Array for

Realtime Investigations of Magnetic Activity (CARISMA), the Canadian High Arctic Ionospheric Network (CHAIN), and the Time History of Events and Macroscale Interactions during Substorms (THEMIS) ground array. Enhancement events can then be correlated with southward propagating TIDs measured across the AM Doppler sounding network in the northeast United States.

Figure 7 shows the AE electrojet index for 0000-1200 UTC on 14 July, 2020, and a map of magnetometer activity at 0330 UTC on the same date. An enhancement in AE occurs at approximately 0340 UTC, 1:50 prior to a southeastward propagating TID with a measured phase velocity of 358 m/s on 14 July, 2020 (Table 2). Assuming that the group and phase velocities are on the same order, the distance traveled by the TID over the 1:50 between the AE enhancement and its arrival

at the AM receiver network is approximately 2360 km. The distance from epicenter of the magnetic activity (Churchill,

Manitoba) to the center of our network (Schenectady, NY) is 2300 km, qualitatively confirming the hypothesis that auroral activity and the southward TIDs are correlated in this case.

Of the thirteen TID events listed in Table 2, seven were associated with auroral activity indicated by enhanced AE indices; these occurred on: 23/03/20, 06/05/20, 14/07/20, 26/09/20, 27/09/20, 29/09/20, and 26/01/21. For four of these, spatial and

temporal correlations between the enhancements of the AE index and the detections of the TIDs, as described above and in Figure 7, were consistent with an auroral cause for the observed TIDs; these occurred on: 06/05/20, 14/07/20, 26/09/20, and 27/09/20. No enhancement in AE index was noted in association with the other TID events in Table 1. The far-right column of Table 3 lists the three-hour Kp index at the time of each TID event. These vary widely from 0-4. Not surprisingly, the seven events associated with auroral activity monitored by AE also are correspond to the highest Kp values.

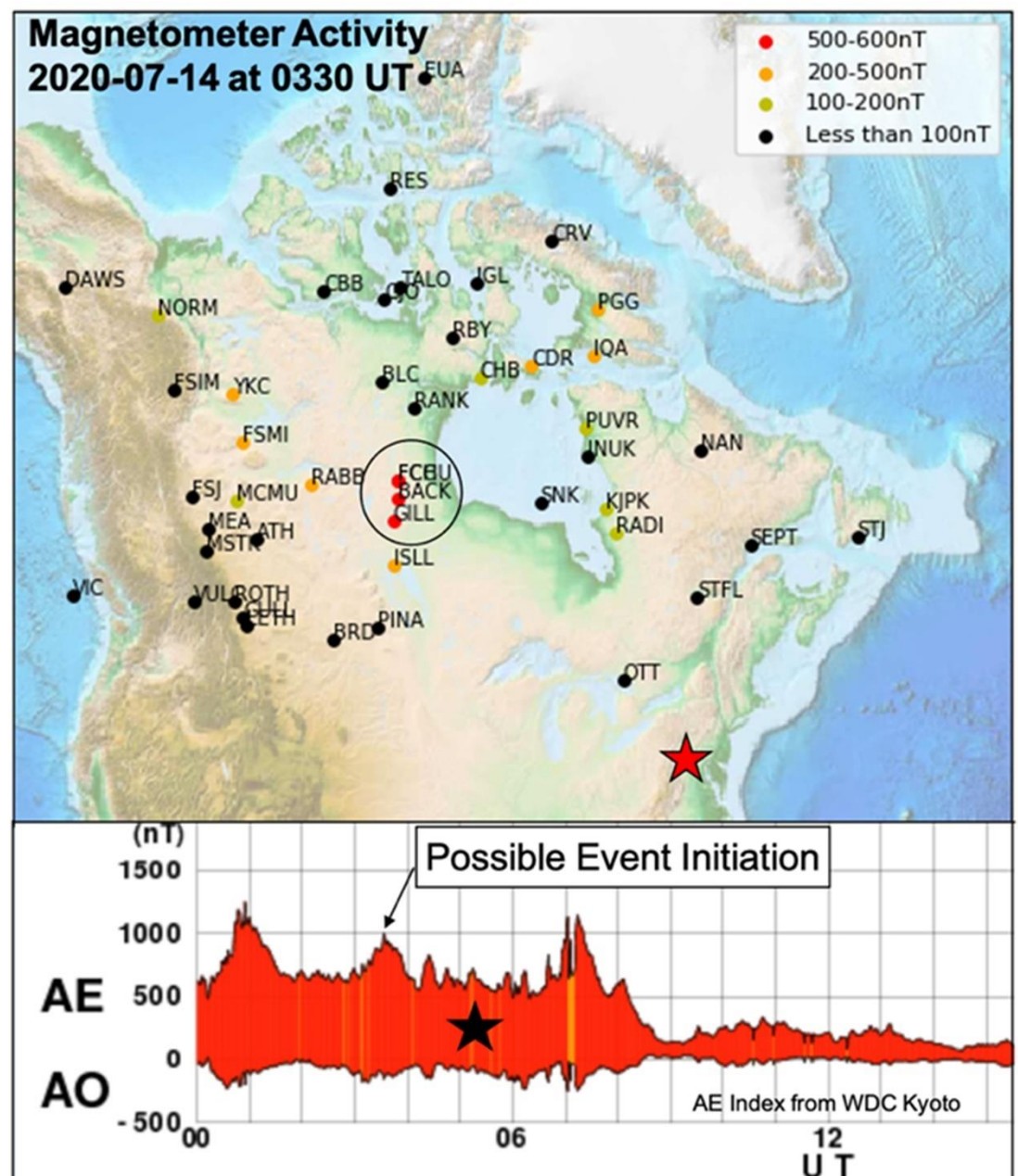

Figure 7: Magnetometer activity (top panel) at 0330 UTC and Auroral Electrojet AE index (bottom panel) for 0000-1500 UTC on 14 July, 2020. at 0330 UTC. The enhancement in AE index at approximately 0340 UTC corresponds to maximum magnetic activity, on the order of 500-600 nT magnitude, in the vicinity of Churchill, Manitoba. This activity precedes a TID event detected with the AM Doppler receiver network in the Northeast United States by 1:50, approximately matching the time it would take a TID with measured the phase velocity of 358 m/s to propagate from Churchill to the measurement array.

## 6. Fully Automated Detection and Characterization of TIDs

The results of Sections 3-5 are based on semi-automatic tracking of variations in Doppler shift on six receivers, supplemented by manual corrections. To enable an operational system that considers larger numbers of events and, more importantly, larger numbers of receivers monitoring larger numbers of clear-channel AM stations, it is essential to remove any manual actions and entirely automate event detection and characterization. To test the efficacy of purely automated analysis, a modified version of the tracking software described in Section 3 was applied to a 12-month data set with

automated initial peak selection and no manual correction.

    Using the resulting Doppler shift versus time on all six receivers tracked fully autonomously, slowness distribution functions were calculated for consecutive 55-minute intervals between sunrise and sunset each day. Intervals for which the maximum slowness correlation value was above an experimentally determined threshold, together with preceding and following

intervals, were re-analyzed with higher effective time resolution to determine the interval of maximum correlation between the six receivers. The phase velocity and direction were calculated for these maximum correlation intervals. Events with multi-modal distribution functions or too-large uncertainties in velocity magnitude or direction were discarded.

    The auto-tracker method was applied to Doppler sounding data from all receiving sites for the three selected clear-channel

AM transmitters for 359 days between 3 April, 2020, and 31 March, 2021. The selected stations were 810 kHz (WGY from Schenectady, New York), 1080 kHz (WTIC from Hartford, Connecticut), and 1560 kHz (WFME from New York City) as shown in Figure 2. The 810 kHz station contained many false positives caused by erroneous slowness correlation coefficients due to the reduced amplitude of the Doppler signatures on WGY, as the lowest frequency station. The false positives were mitigated with a threshold RMS value on the correlation, but event detection still required manual

verification.

    Figure 8 shows the results of the automated event detection. Each triplet of panels represents the set of events detected with a unique transmitting frequency: 810 kHz (top), 1080 kHz (middle), and 1560 kHz (bottom). Within each set, the top two panels show the events as a function of time of day and time of year. For the top two panels in each set, the color bar at right

indicates the magnitude of the phase velocity of each TID. In the second panel, arrows indicate the direction of propagation. In the bottom panel, events are binned by month in histogram format.

## April 2020-March 2021 Automated Event Detection

### 810 kHz

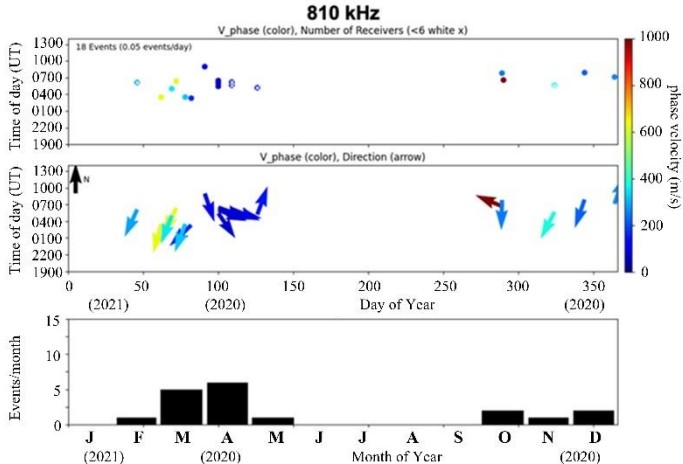

### 1080 kHz

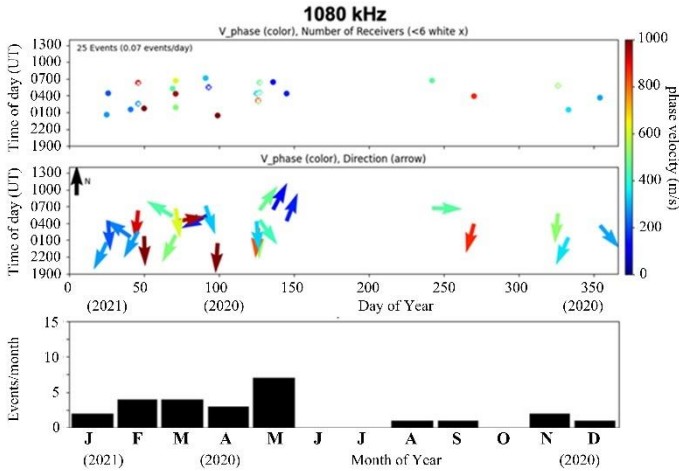

### 1560 kHz

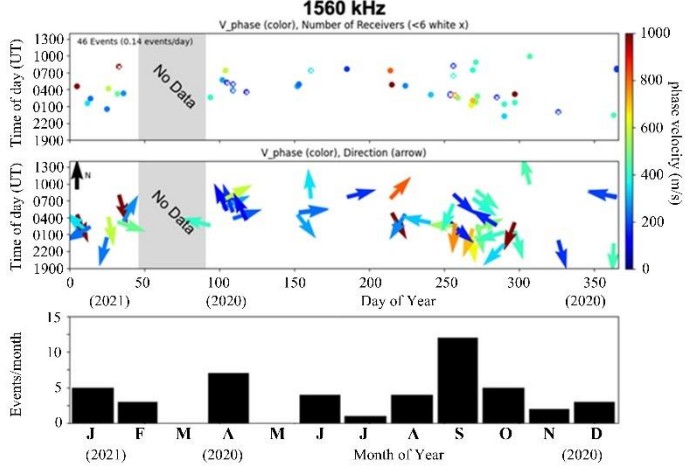

Figure 8: Results of the fully automated tracking and event detection software: Each triplet of panels indicates automatically detected events from 810, 1080, and 1560 kHz AM stations respectively. The top panel of each triplet displays the phase velocity (color) as a function of day of year (x-axis) and time of day (y-axis). The second panel displays the direction of propagation for each event, and the lowest panel groups the detected events by month. Seasonal dependence of maximal event detection suggests the automated tracking favors the detection of LSTIDs (see text discussion). Day of year runs from 1-366, although January-March pertain to 2021 and April-December pertain to 2020.

The method identified 18 TID events on 810 kHz (0.05 events/day), 25 events on 1080 kHz (0.07 events/day), and 46 events on 1560 kHz (0.14 events/day). Significantly, nearly three times the number of events are seen on the highest frequency station compared to the lowest. Manual inspection of the Doppler spectrograms confirms that the variations in the skywave traces are much more distinct and auto-tracking more effective for transmitter higher frequencies > 1 MHz, particularly the variations typical of TIDs. Figure 9 illustrates this difference for a typical TID event observed on 27 September, 2020. The amplitude of the Doppler shift induced when the AM carrier reflects from a vertically moving ionospheric layer is proportional to the frequency of the signal:

$$\Delta f = -2f * \frac{v}{c} \, , \tag{4}$$

where v is the vertical component of the velocity of the ionosphere, $\Delta f$ is the frequency excursion, and f is the carrier frequency. Hence, for given TID parameters, the key measured quantity $\Delta f$ is proportional to the frequency. Furthermore, higher frequencies reflect at higher altitudes where TIDs themselves have greater amplitudes due to energy conversion as the number of neutral particles decreases with altitude (Houghton, 1986). In addition to reduced amplitudes, at lower frequencies it is more common that the radius curvature of the wave is smaller than the mean height. In these case, multiple reflections can occur for the same frequency, and this generates S-shaped features that fold back on themselves (Fišer et al., 2017). Given that the same ionospheric velocities produce larger absolute deviations from the carrier frequency and are less likely to exhibit anomalous features, it is therefore not surprising, as Figure 8 confirms, that the AM Doppler technique for detecting TIDs is much more effective if applied to AM signals in the upper half of the AM frequency band > 1 MHz.

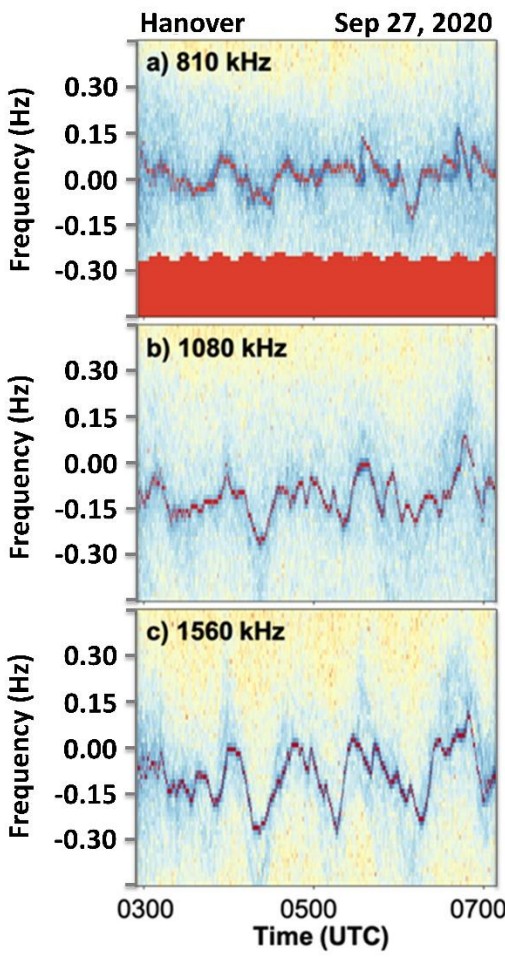

Figure 9: Spectrograms of the 810 kHz, 1080 kHz, and 1560 kHz signals, covering 0.9 Hz bandwidth for 0300-0700 UTC on 27 September, 2020. As expected from equation (4), the Doppler shift is a function of the frequency (larger for higher frequencies). The figure indicates that TIDs may be easier to detect and track using carriers in the upper half of the AM band.

## 7. Seasonal dependence of AM network identified TIDs and comments on technique optimization

Seasonal dependence of TID events identified with the AM Doppler method are shown in the histogram panels of Figure 8. For all three frequencies, there is a distinct minimum occurrence in the summer (days 150-215), as well as hints of equinoctial peaks. The summer peak in occurrence rate is clearly inconsistent with previous studies of nighttime mid-latitude MSTIDs using a variety of techniques. For example, optical imager data recorded at Haleakala from 2006-2012 show winter and summer peaks in occurrence rate of MSTIDs with minima around the equinoxes [Figure 3 of Duly et al., 2013]. Analysis

of two years of optical imager data from two locations in Japan [Figure 4b of Shiokawa et al., 2003] and four years of data at Arecibo [Figure 2 of Martinis et al., 2010] also show winter and summer peaks in occurrence rate. A different technique, GNSS observations from an array of observatories in Europe, shows a winter peak and a weak and a secondary summer peak in occurrence rate of nighttime MSTIDs in the mid-latitude (<55 degrees) portion of the array [Figure 3 of Otsuka et al., 2013]. Another GNSS study with a more global array also finds that the nighttime MSTID occurrence rate peaks in summer in many longitude sectors [Figure 5 of Kotake et al., 2006]. Yet another technique, in situ satellite measurements of F-region plasma density undulations which may be associated with MSTIDs, also reveals equinoctial minima and summer or winter maxima in occurrence rates of nighttime mid-latitude undulations [Figures 2-4 of Park et al., 2010]. Other optical imager studies of mid-latitude nighttime MSTIDs show winter peaks in occurrence rate but are inclusive about summer [Garcia et al., 2000] or show weakest events in Fall and more intense ones in winter and summer [Tsuboi et al., 2023]. The preponderance of all of these previous studies demonstrates that the seasonal dependence found in Figure 8 is not consistent with mid-latitude nighttime MSTIDs.

Possibly the observed seasonal dependence of TIDs measured with the AM Doppler technique is more consistent with mid-latitude nighttime LSTIDs. For example, Tsugawa et al. [2004] observed peaks in LSTID occurrence around the equinoxes in the spring and fall, absence of quiet-time LSTIDs in summer, and minimum occurrence of LSTIDs in summer during active geomagnetic periods. Yakovets et al. [2011] and Ding et al. [2014] detected similar activity peaks at the equinoxes for LSTIDs. Because seasonal statistics of TID characteristics may depend on local time and region, it is difficult to draw firm conclusions, but the seasonal dependence seen in Figure 8 suggests that the automated tracking and event detection algorithm employed here may preferentially detect LSTIDs, perhaps because their larger wavelengths imply that the resulting Doppler shift variation have longer time scales and are easier to track. Another possibility is that the fully automated AM Doppler technique is sensitive to MSTIDs as well as LSTIDs, but that the method is significantly less sensitive in summer, perhaps due to different ionospheric conditions. Detailed interpretation of the seasonal dependence of TIDs detected by the novel fully automated AM Doppler method requires further study of larger data sets.

For all three transmitter frequencies, the detected TIDs primarily propagate southward. Southward components to the phase velocity are observed for 15 of 18 events on 810 kHz, 19 of 25 on 1080 kHz, and 28 of 46 on 1560 kHz. The east-west propagation is split almost equally in all three datasets: 9 out of 18 westward on 810 kHz, 12 out of 25 westward on 1080 kHz, and 22 out of 46 westward on 1560 kHz. On the 810 kHz station, the phase velocities range from 71 to 1308 m/s with a mean phase velocity of 304 m/s and a standard deviation of 295 m/s. On the 1080 kHz station, the phase velocities range from 130 to 1453 m/s with a mean phase velocity of 515 m/s and a standard deviation of 329 m/s. On the 1560 kHz station, the phase velocities range from 104 to 1914 m/s with a mean phase velocity of 445 m/s and a standard deviation of 367 m/s.

Of the 13 events identified for analysis by the manual methods, seven of those events are detected by the auto-tracker on at least one transmitter frequency. The auto-tracker initially detects an additional 2 of the 13 manually detected events, but these two fail to meet the auto-tracker's criteria on the uncertainty in phase velocity magnitude or direction. The imperfect overlap between the manually selected events and those detected with the auto-tracker technique may be attributed to the preferential detection of LSTIDs, the fact that the automated system performs less optimally on the 810 kHz, and the discrepancies that exist between the two frequencies with purely human-in-the-loop detection and tracking.

To investigate discrepancies between the 1560 kHz and the 810 kHz results, the semi-automated analysis described in Section 3 was applied to the 1560 kHz data set for the thirteen TID events for which a similar analysis applied to the 810 kHz data set is summarized in Table 2. In eight out of ten events for which comparison was possible, the direction of propagation of the detected TIDs on the 810 vs 1560 kHz matches within 45°, but the magnitude of the phase velocity on the 1560 kHz station is generally greater than that measured on the 810 kHz station. This discrepancy could be introduced by different reflection heights between the two frequencies, or alternatively it could be due to error introduced because the receivers do not encircle the 1560 kHz station. Therefore, the received signals may be from oblique reflections off the ionosphere, may have different reflection path lengths, and/or may sample a smaller angular spread of the ionosphere. For optimal system designs, receivers should encircle the transmitters equidistant from primary transmitters.

Ultimately, the results of Figure 8 demonstrate the feasibility of automatic TID detection applied to spectral data of Doppler shifts of multiple receiver clear-channel AM signals at multiple receiver locations. The results also indicate this technique is far more effective when applied to signals in the upper part of the AM band. For such signals, the detection software used here has identified approximately one event per week and seasonal statistics suggest the LSTIDs are favored over MSTIDs. Further refinement of the algorithm could result in sensitivity to more events and greater sensitivity to MSTIDs. The automatic event detection demonstrated herein is essential if the technique is to be practical using configurations with large numbers of AM signals measured at a large number of receiving stations.

## 8. Conclusions

This study has demonstrated the efficacy of using a circular network of AM receivers around a single transmitting frequency to identify TIDs and their characteristics from Doppler shifted reflected skywaves. While Chilcote et al. [2015] showed that TIDs can be detected via Doppler sounding using AM signals, their use of only three receivers and a number of different frequencies for a single detected TID did not establish that TID horizontal phase velocity magnitude and direction could be reliably corroborated by other methods, namely GNSS TEC observations. However, a larger sample set with a more robust measurement design from the present study has shown good agreement between horizontal phase velocities determined from GNSS-TEC data and those derived from AM Doppler sounding in slightly more than half of the events studied, suggesting

that these techniques often observe the same TIDs. Differences between the horizontal phase velocities inferred from two data sets in somewhat less than half the cases can be attributed to the different sensitivities of the two measurement techniques in a complex TID wave environment. In this context, it is not surprising that the one example previously studied by Chilcote et al. [2015] showed a significant discrepancy between phase velocities inferred from the two techniques. Most detected TIDs had a southward phase velocity and in four cases, they were associated with enhanced auroral magnetic activity which may have been their source, based on timing, location, and TID velocity.

This experiment, particularly successful implementation of purely automated analysis, clearly indicates that it is possible to extend AM Doppler sounding to large numbers of receivers. Based on these results, a future experiment could be envisioned using a few dozen receivers arrayed around a dozen or more clear channel AM transmitters to achieve continental-scale spatial coverage, allowing TIDs to be tracked for ~1000 km or more. Such an array would effectively complement spatial coverage of the GNSS technique, allowing differences and similarities between the two techniques to be better understood, to the improvement of both. An array covering continental or half-continental scale would also supplement existing optical, digisonde, and SuperDARN radar techniques, further leveraging investments and science yield from those already productive assets. We suggest that future AM Doppler network experiments should employ transmitters in the upper part of the AM band for best results. Additional refinement of the purely automated algorithm may lead to greater sensitivity and more event detections.

**Data Availability**

Data used in this study are available through LaBelle [2024].

**Author Contribution**

CCT performed most of the data analysis and wrote the initial draft of the manuscript with contributions from the other co-authors. JL participated in deploying the instruments, data acquisition, data analysis, and manuscript preparation. PJE participated in deploying instruments to the Schenectady site, data analysis and manuscript preparation. SZE participated in data analysis and manuscript preparation. DM participated in development, construction, and deployment of the radio receiving systems, and consulted on data analysis and manuscript preparation. TK participated in development and maintenance of software for acquisition and archiving of the radio data.

**Competing Interests**

The authors declare that they have no conflict of interest.

## Acknowledgments

The work at Dartmouth College was supported by National Science Foundation grant AGS-1915058. Total electron content data is provided to the community through Millstone Hill Geospace Facility operations at MIT Haystack Observatory under National Science Foundation grant AGS-1952737 to the Massachusetts Institute of Technology. SRZ acknowledge NSF award AGS-2149698 and NASA grant 80NSSC21K1310. The authors thank many Dartmouth undergraduate students who contributed to various stages of this project, including Rongfei Lu, Jeffrey Kim, Eric Tao, Eldred Lee, Nathan Grice, and Carter

Person. The authors acknowledge the citizen science contributions of the late John S. Erickson, who provided internet and station infrastructure to this project at his Schenectady, NY residence for the crucial WGY reference clear channel station recording.

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
