# Peer review of "Tracking Traveling Ionospheric Disturbances through Dopplershifted AM radio transmissions"

_EGUsphere, 2024_

## Referee Comment (RC1)

Comments on **"Tracking Traveling Ionospheric Disturbances through Doppler Shifted AM Radio Transmissions"** by Trop et al.

**General Comments:**

This study investigates Traveling Ionospheric Disturbances (TIDs), focusing particularly on nighttime TIDs, using a group of AM radio receivers and transmitters. Notably, the authors proposed an automated technique to track these disturbances, which serves as an effective tool for analyzing large datasets. Overall, the study's objective is intriguing, and the manuscript is well-written. However, several key points have not been addressed. For example, the authors used a radio frequency below 2 MHz, which can be significantly influenced by E region electron density. Yet, they did not present the E region conditions during the study period (April 2020 - March 2021). While it's understandable that nighttime data was selected to mitigate E region effects, nighttime sporadic E layers, which are common in this frequency range, pose substantial challenges for this study. Since AM radio waves reflect from the lower ionosphere, it would be beneficial to compare the results with OI630nm airglow imager data rather than rely solely on dTEC data (which could be used as a complementary source).Additionally, the seasonal characteristics of TIDs seem unusual. Typically, receivers located within a 200 km radius are more effective at identifying Medium-Scale TIDs (MSTIDs) than Large-Scale TIDs (LSTIDs) because MSTIDs cause greater density fluctuations over shorter distances, resulting in a higher electron density gradient. I recommend that the editor consider a major revision. Detailed comments follow:

**Specific Comments:**

1. It is crucial to provide information about the E region conditions during all events. For the thirteen events listed in Table 2, please check the Es layer conditions and compare the results with co-located airglow observations (based on availability).

2. In Table 3, many events show more than a 100 m/s phase velocity difference between the AM radio wave-detected TIDs and those observed via dTEC. The reasons for this discrepancy should be addressed more thoroughly by using the airglow observation as a complement.

3. As pointed out by Chilcote et al. (2015), the time window of the dTEC estimation method may not impact the phase velocity of TIDs. This can be verified by testing different time windows (such as 30 minutes or 1 hour).

4. Figure 8 is confusing. The title indicates the timeframe of April 2020 - March 2021, but the X-axis is labeled in days of the year starting from zero. Additionally, the titles for panels 8c, g, and j indicate events per month while the X-axis is in days; this inconsistency needs clarification.

5. As mentioned in the general comments, the results regarding seasonal variation appear strange, especially during summer and winter when MSTID activity is heightened over the U.S. sector. Please double-check these results against airglow imager or ionosonde observations.

**Minor Comments:**

- Line 33: Fritts and Alexander (2003) did not focus on gravity waves in the ionosphere/thermosphere.

- The papers by Cosgroves and Tsunoda (2000, 2006, etc.) are relevant and should be cited in line 34.

- According to line 81, sections 2 and 3 should be titled "Instrumentation" and "Methodology," respectively.

- Line 159 should include details on relevant local time.

- Line 167: Provide more detailed information about the semi-automated tracking method used in this study, including how manual corrections are conducted.

- Line 319: Correct "Narayan" to "Narayanan."

- Please verify the geomagnetic conditions for all events listed in Tables 2 and 3.

---

## Author Response (AR1)

Response to referees of "Tracking Traveling Ionospheric Disturbances through Doppler Shifted AM Radio Transmissions" by Trop et al.

We thank both referees for their careful reading of the manuscript and their helpful suggestions which have resulted in significant improvements of the paper, which we believe is now suitable for publication. Below find a detailed response to each point raised by each of the referees. One of the principal changes is modification of the conclusions somewhat regarding the comparison of horizontal phase velocities measured at the same times by the AM Doppler technique and the GNSS-TEC technique. The responses have also entailed adding many citations to the reference list.

Responses to the five enumerated comments of Referee 1:

1. *"It is crucial to provide information about the E region conditions during all events. For the thirteen events listed in Table 2, please check the Es layer conditions and compare the results with co-located airglow observations"*

We examined vertical ionograms at 7.5 minute cadence from mid-latitude observations at Westford, MA (42.6 N latitude / 288.5 E longitude). The ionosonde data, comprising station MHJ45 within the Global Ionospheric Radio Observatory (GIRO), originated from the University of Massachusetts Lowell's ionosonde station operated on the MIT Haystack observatory grounds. MHJ45 was located on the eastern edge of the study region. No sporadic E was recorded during any of the TID events in Tables 2-3 of our paper. Around event times, Sporadic E was detected about an hour before one of the events (on Sept 26, 2020) and an extremely weak Sporadic E layer was detected at 2 MHz frequency for 90 minutes after one of the events (December 30, 2020). Based on this survey, we conclude that our study was not significantly impacted by sporadic E propagation. This information and conclusion is described in text added at lines 311-318.

2. *"In Table 3, many events show more than a 100 m/s phase velocity difference between the AM radio wave-detected TIDs and those observed via dTEC. The reasons for this discrepancy should be addressed more thoroughly by using the airglow observation as a complement."*

We have done as the referee suggests, but this comment is closely linked to comment 3 of referee 1; see below.

3. *"As pointed out by Chilcote et al. (2015), the time window of the dTEC estimation method may not impact the phase velocity of TIDs. This can be verified by testing different time windows (such as 30 minutes or 1 hour)."*

We have re-run the analysis of the GNSS-TEC data using a 60-minute detrending window to complement the former analysis using a 30-minute window. A column has been added to Table 3 which now shows both results. In seven cases the change in detrending window makes a significant difference in the inferred horizontal phase velocity. This is discussed in the paper at lines 350-onward.

Based on these results, we have replaced and added significant discussion of the comparison between the two techniques at lines 359-375. The significant change is that we have altered the conclusions of the paper somewhat: a slight majority of the events studied show pretty good agreement by our criteria between horizontal phase velocities measured with GNSS-TEC versus those measured with AM Doppler sounding, and a slight minority of events show significant differences in either direction, magnitude, or both. We believe that the reason for these differences lies in the complexity of the wave environment a fair fraction of the time, which includes presence of multiple waves that can be detected with different efficiencies by the different methods.

Accordingly, we have changed wording in the conclusions at lines 611-617 and in the abstract at lines 18-19.

4. *"Figure 8 is confusing. The title indicates the timeframe of April 2020 - March 2021, but the X-axis is labeled in days of the year starting from zero. Additionally, the titles for panels 8c, g, and j indicate events per month while the X-axis is in days; this inconsistency needs clarification."*

We have significantly re-worked Figure 8, changing the labels (and improving the font size in the process). We have also added a sentence to the figure caption.

5. *"As mentioned in the general comments, the results regarding seasonal variation appear strange, especially during summer and winter when MSTID activity is heightened over the U.S. sector. Please double-check these results against airglow imager or ionosonde observations."*

We have done a more thorough literature search, finding more papers than we previously cited which address the seasonal dependence of mid-latitude nighttime MSTIDs using a range of techniques, including GNSS-TEC, in-situ satellite, and optical imaging as suggested by the referee. These papers, reviewed in text added at lines 539-555, establish a consistent and robust pattern of seasonal dependence which differs significantly with that discovered with the fully automated version of the AM Doppler technique. Discussion of this discrepancy follows in a paragraph added at lines 557-567.

Responses to the minor comments of Referee 1:

*"Fritts and Alexander (2003) did not focus on gravity waves in the ionosphere/thermosphere; The papers by Cosgroves and Tsunoda (2000, 2006, etc.) are relevant and should be cited in line 34; According to line 81, sections 2 and 3 should be titled Instrumentation and Methodology, respectively; Line 159 should include details on relevant local time; Line 167: Provide more detailed information about the semi-automated tracking method used in this study, including how manual corrections are conducted; Line 319: Correct Narayan to Narayanan; Please verify the geomagnetic conditions for all events listed in Tables 2 and 3."*

We have addressed all of the minor comments raised by Referee 1. Providing more information about the semi-automated tracking entails significantly enhanced longer text inserted at lines 179-187. Verifying geomagnetic conditions entails adding a column to Table 3 with the Kp index associated with each event, and added discussion at lines 449-455. We replaced reference to Fritts and Alexander (2003) with reference to a different review paper.

Responses to the five enumerated comments of Referee 2:

1. *"Three methods are used to estimate TID velocity. However, I had a hard time understanding the sine method..."*

We replaced the paragraph describing the "sine method" with a lengthier more more precise mathematical description at lines 224-240, along with an alteration to Figure 2.

2. *"Why didn't you compare between the methods for TID estimation and present that in table 2..."*

We have added columns to Table 2 which give horizontal phase velocity direcions and magnitudes inferred from the triad and sine methods, to complement the values given previously for the slowness method. We have altered and added discussion of these results at lines 279-287.

3. *"How were your uncertainties calculated as presented in Table 2? I didn't see anything in the paper regarding this point. The uncertainty estimates are also fairly sizable..."*

We have modified the description of the uncertainty calculation at lines 260-264. As noted, the relative uncertainty values in the table are useful for comparing one phase velocity to another calculated with the slowness method, but they do not convey the absolute uncertainty of the measurement.

4. *"I think a limitation of your investigation is that the baselines are fairly short. This should be discussed..."*

We added sentences at lines 134-137 justifying the choice of baselines in this experiment.

5. *"It wasn't clear to me but were three always days of large AE that corresponded to the LSITDs? Or were there intervals when it was geomagnetically quiet, but you still say LSTIDs?"*

We have altered and added discussion of the auroral activity associated with the events at lines 449-455, including specifying which events were not associated with auroral activity indicated by AE index connected to the events via appropriate timing. We have added a column in Table 3 providing the 3-hour Kp index for each event.